# Roles of the GA-mediated *SPL* Gene Family and miR156 in the Floral Development of Chinese Chestnut (*Castanea mollissima*)

**DOI:** 10.3390/ijms20071577

**Published:** 2019-03-29

**Authors:** Guosong Chen, Jingtong Li, Yang Liu, Qing Zhang, Yuerong Gao, Kefeng Fang, Qingqin Cao, Ling Qin, Yu Xing

**Affiliations:** 1Beijing Advanced Innovation Center for Tree Breeding by Molecular Design, Beijing University of Agriculture, Beijing 102206, China; cgschennuo@163.com (G.C.); wangyi1314aa@163.com (J.L.); bualiuyang@163.com (Y.L.); gaoyr@cau.edu.cn (Y.G.); 2College of Plant Science and Technology, Beijing Key Laboratory for Agricultural Application and New Technique, Beijing University of Agriculture, Beijing 102206, China; zhangqing@bua.edu.cn; 3Beijing Collaborative Innovation Center for Eco-environmental Improvement with Forestry and Fruit Trees, Beijing 102206, China; fangkefeng@126.com; 4College of Biological Science and Engineering, Key Laboratory of Urban Agriculture (North China), Ministry of Agriculture, Beijing 102206, China; caoqingqin@bua.edu.cn

**Keywords:** *Castanea mollissima*, miR156, *SPL* gene family, GA, floral development, expression patterns

## Abstract

Chestnut (*Castanea mollissima*) is a deciduous tree species with major economic and ecological value that is widely used in the study of floral development in woody plants due its monoecious and out-of-proportion characteristics. Squamosa promoter-binding protein-like (*SPL*) is a plant-specific transcription factor that plays an important role in floral development. In this study, a total of 18 *SPL* genes were identified in the chestnut genome, of which 10 *SPL* genes have complementary regions of *CmmiR156*. An analysis of the phylogenetic tree of the squamosa promoter-binding protein (SBP) domains of the *SPL* genes of *Arabidopsis thaliana*, *Populus trichocarpa*, and *C. mollissima* divided these *SPL* genes into eight groups. The evolutionary relationship between poplar and chestnut in the same group was similar. A structural analysis of the protein-coding regions (CDSs) showed that the domains have the main function of SBP domains and that other domains also play an important role in determining gene function. The expression patterns of *CmmiR156* and *CmSPLs* in different floral organs of chestnut were analyzed by real-time quantitative PCR. Some *CmSPLs* with similar structural patterns showed similar expression patterns, indicating that the gene structures determine the synergy of the gene functions. The application of gibberellin (GA) and its inhibitor (Paclobutrazol, PP_333_) to chestnut trees revealed that these exert a significant effect on the number and length of the male and female chestnut flowers. GA treatment significantly increased *CmmiR156* expression and thus significantly decreased the expression of its target gene, *CmSPL6*/*CmSPL9*/*CmSPL16*, during floral bud development. This finding indicates that GA might indirectly affect the expression of some of the *SPL* target genes through miR156. In addition, RNA ligase-mediated rapid amplification of the 5′ cDNA ends (RLM-RACE) experiments revealed that *CmmiR156* cleaves *CmSPL9* and *CmSPL16* at the 10th and 12th bases of the complementary region. These results laid an important foundation for further study of the biological function of *CmSPLs* in the floral development of *C. mollissima*.

## 1. Introduction

Chinese chestnut (*Castanea mollissima* BL.) is rich in nutrients, resistant to disease, and barren, and these features make Chinese chestnut an important ecological and economic tree species. As a monoecious plant, the proportion of female and male flowers in Chinese chestnut can reach 1:2400–4000. A large number of male flowers consume too much tree nutrition, which is one of the important reasons for the low yield of chestnut [1,2]. Therefore, exploration of the mechanism of floral development is important for regulating the ratio of female to male flowers, which might provide valuable information for solving the problem of the low chestnut yield. Floral development is mainly influenced by genetic factors and environmental conditions [3], and phytohormones have been reported to play an important role in floral development [4]. For example, in floral regulatory networks, gibberellins (GAs) are critical for the development of reproductive organs, especially for floral sex determination [5,6]. Auxin determines whether the primordium of the flower is formed during the initial process of the organ and the specificity of the flower organ formed during organogenesis [7]. Cytokinins (CKs) contribute to the sex control of *Jatropha curcas* [8] and have been used for many years in horticultural production practices [9]. Genes related to phytohormone biosynthesis and signaling have been identified and are reportedly involved in floral development in several species [10]. A naturally occurring male inflorescence mutation was found on a branch of a Chinese chestnut tree in the mountains near Beijing, China, with a male inflorescence length reduced from 1/6 to 1/8 of the wild-type male inflorescence on the same tree, and the application of exogenous GA_3_ partially restored the sck1 phenotype to the wild-type phenotype [2]. Therefore, it is very important to investigate the effect of GA on floral development in chestnut by spraying the whole tree with GA and its biosynthesis antagonist, PP_333_.

Transcription factors (TFs) precisely coordinate gene expression by activating or inhibiting transcription in response to various endogenous and environmental signals and are usually divided into different families based on the sequences of the DNA-binding domain and other conserved motifs [11]. Plants have thousands of TF families, such as *MYB*, *NAC* (*NAM*, *ATAF1/2*, and *CUC2*), auxin response factors (*ARFs*), Squamosa promoter-binding protein-like (*SPL*), and *bHLH*, which play crucial roles in the regulation of the gene networks involved in many important developmental processes and defense responses in plants [12,13,14,15,16]. Among these TF families, the *SPL* family members are important TFs in plant floral development and have been identified in *Arabidopsis* [17], *Populus trichocarpa* [18], *Betula luminifera* [19], *Brassica napus* [20], *Petunia hybrida* [21], and other flowering species. *SPLs* are plant-specific TFs that contain a highly conserved SBP binding domain of approximately 80 amino acids, which contains two zinc finger structures and a nuclear localization signal [22,23]. Based on the important role of the *SPL* gene family in plant floral development, ranging from vegetative to reproductive growth [24], male sterility [25], GA synthesis [26], and floral morphogenesis [27], researchers have identified *SPL* genes in a growing number of species: 16 *SPL* genes in *Arabidopsis* [17], 28 *SPL* genes in *P. trichocarpa* [18], 21 *SPL* genes in *P. hybrida* [21], and 18 *SPL* genes in *B. luminifera* [19].

MicroRNAs (miRNAs) are approximately 21–23 nucleotides (nt) long, small single-strand noncoding RNAs which are encoded by endogenous genes. miRNA plays a key role in regulating gene activity at post-transcriptional levels by binding to complementary sites in target genes, causing degradation and/or translational repression of the target mRNAs [28]. miRNA functions broadly to regulate many aspects of plant development. For instance, miR160 or miR167 targets *ARFs*, which are involved in auxin or defense responses or play roles at many stages of development [14,29]; *MYB* TFs are antagonized by miR159 in the control of floral development and heat response or in conjunction with miR828 to modulate fiber development [12,30]; and *NAC* TFs, which are encoded by miR164 target genes, play roles in lateral root development, boundary specification, drought tolerance, or the immune response [13,31,32]. The majority of *SPL* genes in many species have complementary regions of miR156, and the miR156/*SPL* mode of action has been shown to play a key regulatory role in many growth and development processes [33,34,35,36]. For instance, either the overexpression of csi-miR156a or the individual knockdown of one of its two target genes, *CsSPL3* and *CsSPL14*, significantly enhances the somatic embryogenesis competence of citrus callus [37]. *PvSPL4* is the target gene of the microRNA pvi-miR156 and controls the initiation of aerial axillary buds of switchgrass [38]. miR156 regulates *Arabidopsis* pollen production, fertility, and elongation of floral organs (including petals, sepals, and wafers) by inhibiting the expression of the target gene *SPL2* [27].

The floral development of chestnut shows differences from those of *Arabidopsis* and *P. trichocarpa*, which are characterized by monoecism (with unisexual flowers) and large differences between male and female flowers. Therefore, it is important to study the genes related to floral development in chestnut. In this study, to systematically analyze the characteristics of the *SPL* genes in chestnut, the molecular structures and expression patterns of *SPL* genes in *C. mollissima* were compared with those of *Arabidopsis* and *P. trichocarpa*, and the relationship between *SPL* genes among different species and the role of *CmSPL* genes in expression regulation were examined. In addition, the effects of GA on both male and female flowers of *C. mollissima* were investigated by treating chestnut trees with GA, and the expression patterns of *CmSPL* and *CmmiR156* during floral bud development and the GA-mediated induction of the expression of these genes were explored.

## 2. Results

### 2.1. Effects of GA and Its Inhibitor on the Number of Male and Female Inflorescences

To analyze the effects of GA on floral development, GA and its inhibitor were applied to 18-year-old trees. Because the chestnut female flower undergoes differentiation outside of the bud and floral morphogenesis occurs after germination, GA and PP_333_ were first sprayed before the floral buds germinated in late March. The first sampling was performed once the buds turned green, and both GA and PP_333_ spraying and sampling were performed at 1-week intervals until the male inflorescence appeared, at which time sampling was stopped. During this period, a total of four samples, denoted Floral Bud 1 (FB1)–FB4, were obtained, and the floral bud morphology was observed, as shown in Figure 1A. The number and lengths of male and female flowers were investigated at the flowering stage, and the results are shown in Table 1. Treatment with 100 mg/L of GA increased the number of male inflorescences on each branch and significantly reduced the ratio of female flower clusters to male inflorescences, the male inflorescence length, and the mixed inflorescence length. The GA treatment also decreased, albeit not significantly, the number of female flowers. In contrast, treatment with the GA inhibitor PP_333_ at a concentration of 1000 mg/L significantly increased the number of female flowers and insignificantly increased the mixed inflorescence length.

To analyze the effects of the exogenous addition of the hormone GA and the GA inhibitor PP_333_ on endogenous hormones, the expression levels of key genes in the GA synthesis and signaling pathways were determined (Figure 1B). Three key genes in in the GA synthetic pathway, namely, *ent-kaurene synthase* (*KS*), *ent-kaurene oxidase* (*KO*), and *ent-kaurenoic acid oxidase* (*KAO*), and two typical genes in the GA signaling pathway, namely, *DELLA* and *GA-insensitive drawf1* (*GID1*), were selected [39]. Their analysis revealed that *KS* and *GID1* gene expression showed a decreasing trend during floral bud differentiation, and no significant difference was obtained after the treatments. During PP_333_ treatment, *KO*, *KAO*, and *DELLA* expression was significantly decreased during the first floral bud stage, and the overall trend was downward. No significant differences were found between the control and GA treatments.

Some genes play important roles in flower development [15,40,41], and six of these (*SPL9*, *SPL16*, *SOC1*, *AP1*, *FUL*, and *LEY*) were selected. The gene expression levels in floral buds at different developmental stages were determined by fluorescence quantitative RT-PCR analysis, and the expression trends of the six genes were different. Specifically, *SPL9*, *SPL16*, and *AP1* showed an upward trend during floral bud development, and the *SOC1*, *FUL*, and *LEY* genes showed a downward trend. After GA treatment, the expression levels of *SPL9* and *SPL16* were significantly decreased, whereas no significant differences in the gene expression levels of *AP1*, *FUL*, and *LEY* were found between the GA-treated and control floral buds (Figure 2). These results suggested that *SPL9* and *SPL16* might be involved in not only flower development but also the GA signaling pathway in chestnut.

### 2.2. Identification of *CmSPL* Genes

To further explore the important role of the *SPL* genes in flower development, we analyzed the cDNA sequence of the *SPL* gene family in chestnut and included the deduced protein length, molecular weight, isoelectric point, and aliphatic index. The deduced length of the CmSPL proteins ranged from 151 (*CmSPL15*) to 1091 (*CmSPL14*) amino acids, they had a predicted molecular mass of 17.3 (*CmSPL15*) to 120.6 (*CmSPL14*) KDa, the pI values ranged from 5.72 (*CmSPL7*) to 9.39 (*CmSPL2*), and the Ai varied from 29.74 (*CmSPL15*) to 84.41 (*CmSPL1*). In addition, 10 of the 18 genes belonging to the *SPL* family have a target site for *CmmiR156*, and two types of complementary regions of *CmmiR156* are found in these *CmSPLs*, most of which are located in the coding region and a few are located in the 3’ UTR region of *CmSPL* (Table 2). These results indicated the diversity in the features of the *CmSPLs* in *C. mollissima*.

### 2.3. Phylogenetic Analysis of *SPL* Genes

The *SPL* gene function has been almost fully clarified in *Arabidopsis* due to its importance as a model plant. To further study the evolutionary relationship among *C. mollissima*, *Arabidopsis*, and *P. trichocarpa*, the sequences of 18 *CmSPLs* from chestnut, 13 *AtSPLs* from *Arabidopsis*, and 28 *PtSPLs* from *P. trichocarpa* were analyzed through a phylogenetic tree analysis. All 59 *SPL* genes were divided into eight groups, and seven of the groups (all except IV) contained at least one *SPL* gene from three species (Figure 3). V includes the smallest *SPL* gene, which contains less than 210 amino acids, and VII encodes a large set of *SPL* genes with approximately 1000 amino acids (Figure 3 and Table 2). Therefore, although the construction of the phylogenetic tree was based on the SBP domain, the tree also reflected the evolution and classification of the full-length *SPL* genes, which also indicated that the SBP domain is a relatively conserved sequence among all *SPL* genes of different species. In addition, with the exception of *CmSPL17* in I, the homologous relationship between other chestnut *SPL* genes and the poplar *SPL* gene is closer than that of chestnut genes with *Arabidopsis*, which indicates that the conservation of the *SPL* gene family among woody plants is higher than that among woody and herbaceous plants.

### 2.4. Identification of Conserved Motifs in *CmSPLs*

The detailed domain structure of *CmSPL* was analyzed by NCBI database alignment, and the results showed that the SBP domain is the only conserved domain shared by all *CmSPLs*. As shown in Figure 4A, all SBP domains in *CmSPLs* also contain two zinc finger structures (Zn-1/2) and a conserved nuclear localization signal (NLS). The Zn-1 motif is a CCCH (C3H) type in all CmSPL proteins except that the His residue in *CmSPL7* is substituted with a Cys residue. Unlike Zn-1, the characteristic sequence of Zn-2 (C2HC) is highly conserved in all *CmSPLs*, which partially overlap NLS.

In addition to the SBP domain, other conserved domains also play an important role in protein function. For example, ANK domains involved in protein–protein interactions were relatively conserved in most of the investigated proteins. An analysis of 10 conserved domains in 18 *CmSPLs* was performed using the MEME web server (Figure 4B). Consistent with the phylogenetic tree analysis, we also divided the structures of these 31 SPL proteins into eight groups. Most of the domains were relatively conserved between *Arabidopsis* and chestnut and were also relatively conserved in poplar trees. The number of motif(s) in each *CmSPL* varied from 1 to 9, and most *CmSPLs* within the same group shared similar motif profiles (Figure 4B). Among these motifs, motif 1 is the SBP domain, which exists in all *CmSPLs* analyzed, and motif 2 consists of a sequence containing a region complementary to miR156.

### 2.5. Expression Profiles of the *SPL* Genes and miR156 in Organs of *C. mollissima*

Both the evolutionary relationship and gene structure analyses showed the conservation and diversity of *SPL* genes. We also aimed to explore the role of these *SPL* genes in the development of Chinese chestnut flowers and thus analyzed the expression patterns of 18 genes in different organ samples (female flowers, male flowers, leaves, ovary, stigma, and stamen) by qRT-PCR. In general, the *CmSPL* genes were classified into two types based on their expression profiles (Figure 5B,C). A minority of the *CmSPL* genes, namely, *CmSPL1*, *CmSPL12*, and *CmSPL14*, showed relatively high expression in all the examined organs (Figure 5B). All of these genes are devoid of miR156 complementary regions and are therefore referred to as the *CmmiR156*-nontargeted *CmSPL* genes. The remaining *CmSPL* genes, most of which are *CmSPL* genes with complementary regions of *CmmiR156*, are therefore known as *CmmiR156*-targeted *CmSPL* genes, showing more differentiated expression patterns in different organs (Figure 5). This difference implies that *CmmiR156*-targeted and -nontargeted *CmSPL* genes play distinct roles in flower development in *C. mollissima*. It is worth noting that similar expression patterns were observed in genes belonging to the same group, such as *CmSPL5/13/17* in group I, *CmSPL1/12/14* in group VII, and *CmSPL2/15* in group V (Figure 5). This suggests that the same group of *CmSPLs* plays a similar function in flower development due to their identical motifs. Furthermore, we also studied *CmmiR156* transcript levels in different organs of *C. mollissima* by qRT-PCR. According to its expression pattern, *CmmiR156* was mainly expressed in the male floral cluster and stamen, and the level of *CmmiR156* in these organs was approximately twofold higher than that found in the female flower, which presented the third highest expression of this gene (Figure 5A). In contrast, most of the *CmmiR156*-targeted *CmSPL* genes showed higher expression in male flowers than in male floral clusters and stamen.

### 2.6. Expression Pattern of miR156 and Its Target Genes in Floral Buds at Different Stages under GA Treatment

GA is an important hormone in plant growth and development and plays an important role in floral transition. In our study, *CmmiR156* was downregulated in the development of floral buds and increased significantly after GA treatment (Figure 6A). Thus, the expression patterns of 18 *CmSPL* genes during floral bud development were also analyzed. Most of the *CmmiR156*-targeted *CmSPL* genes, particularly *CmSPL6*, *CmSPL9*, *CmSPL10*, *CmSPL13*, and *CmSPL16*, were substantially upregulated during floral bud development (Figure 6B). In addition, *CmSPL6*, *CmSPL9*, and *CmSPL16* expression was significantly decreased during GA treatment and showed the opposite expression patterns from that of *CmmiR156*. In addition, the *CmmiR156*-nontargeted *CmSPL* genes exhibited a constitutive expression pattern during floral bud development, and the expression level did not significantly differ under GA treatment (Figure 6C). These results demonstrate that *CmmiR156*-targeted and -nontargeted *CmSPLs* play an important role in the floral bud development of *C. mollissima*, and there is a functional difference between them.

### 2.7. Validation of *CmmiR156*-Targeted *CmSPLs* Cleavage Sites by RNA Ligase-Mediated Rapid Amplification of the 5′ cDNA Ends (RLM-RACE)

In our study, 10 *CmSPL* genes with complementary regions for *CmmiR156* were identified. Most of the CRs were highly paired with only one base mismatch located at the eighth base (Figure 7A). The *CmSPL9* and *CmSPL16* target genes were selected for section analysis using the RLM-RACE technique. Cloning and sequencing of the PCR amplicons of remnant mRNAs enabled the determination of the nucleotide position in which a slicing event occurred. *CmmiR156* sliced its *CmSPL9* and *CmSPL6* target genes at different sites with different efficiencies. Specifically, the *CmSPL9* and *CmSPL16* slicing events occurred at the 12th/10th nucleotide in the CR, and the efficiencies of these splicing reactions were 10/10 and 8/10, respectively (Figure 7B).

## 3. Discussion

*SPLs* are plant-specific transcription factors that contain a highly conserved SBP domain and have a large family in plants. Due to the important function of *SPL* genes in plant development, more and more species of *SPLs* have been identified. Chestnut is an important ecological and economic crop, and its *SPL* genes have not been systematically studied. Because of the different proportions of male and female flowers in chestnut and the important function of *SPL* genes in floral development, we believe that the identification and analysis of chestnut *SPL* genes is of great significance. In this study, we identified 18 *SPL* genes in the *C. mollissima* genome, and expression analysis showed that most of them had relatively high expression in floral organs (Figure 5). Compared with *Arabidopsis thaliana* [17], *Vitis vinifera* [42], *P. hybrida* [21], *P. trichocarpa* [18], and *B. luminifera* [19], the number of chestnut *SPL* genes is similar to that of *Arabidopsis* (16), *B. luminifera* (18) and *V. vinifera* (18), but notably lower that of *P. trichocarpa* (28), and *P. hybrida* (21). This finding means that the *SPL* genes of different plants evolve in a species-specific manner, which may be affected by different genetic repeat events.

According to the phylogenetic tree analysis, we divided 18 *CmSPL* genes into eight groups (I–VIII) (Figure 3). The tight integration of the *CmSPL* genes with the *AtSPL* and *PtSPL* genes suggests that the first origins of these *SPL* genes may be identical and may have a common ancestor. By conservative motif analysis, it was found that all *CmSPL* genes from the same phylogenetic group had the same motif (Figure 4B), indicating that the genes of the same phylogenetic group may have similar functions in *C. mollissima*. To investigate the potential role of *CmSPLs* in the floral development of chestnut, the expression profiles of *CmSPL* genes in six different organs of chestnut were analyzed. In addition, due to the correlation between *CmSPLs* and *CmmiR156*, we also analyzed the expression profile of *CmmiR156*. The expression pattern of the *CmSPL* genes in different organs of chestnut are shown in Figure 5. Some *CmSPLs* (*CmSPL1*, *CmSPL12*, and *CmSPL14*) exhibited a constitutive expression pattern in the examined organs, and the remaining genes exhibited development- and organ-dependent expression patterns. Notably, the constitutive expression patterns were typically observed in the *CmmiR156*-nontargeted *CmSPL* genes, whereas more than half of the *CmmiR156*-targeted *CmSPLs* tended to exhibit different expression patterns. The miR156-targeted and -nontargeted *SPL* genes also have similar expression differences in *P. trichocarpa*, *B. luminifera*, *B. napus*, and *P. hybrida* [18,19,20,21]. This finding suggests that miR156-targeted and -nontargeted *SPLs* undergo different evolutionary patterns of differentiation expression, suggesting that they play distinct roles in the development and growth of plants.

The DELLA protein has been found to interact with the SPL protein as a central transcriptional repressor of the GA response and to inhibit SPL protein activity. The physical interaction between SPL and DELLA is considered to be an integrator of age and GA-related pathways at flowering [43]. This experiment revealed no significant difference in the level of DELLA expression under GA treatment compared with the control treatment, which might indicate that DELLA is not sensitive to GA. The first spraying treatment with PP_333_ significantly reduced the expression of DELLA (Figure 1B). This protein is also involved in the transmission of various plant hormones and environmental signals, such as abscisic acid (ABA) [44], jasmonic acid (JA) [45], sugar signal [46], and light signal [47,48]. The reduced DELLA expression level might be due to the fact that the tree just started to come into contact with PP_333_, resulting in an imbalance in other internal hormones, which in turn affects the expression level of DELLA protein. The extensive interaction between DELLA protein and miR156-targeted *SPLs* has been demonstrated in several species by yeast two-hybrid assays. For example, in *B. luminifera*, *BlRGA* can interact with *BlSPL1*, *BlSPL6*, *BlSPL8*, *BlSPL13*, and *BlSPL18*, and *BlRGL* can interact with *BlSPL1*, *BlSPL8*, *BlSPL13*, *BlSPL16*, and *BlSPL18* [19]. In this study, we found that the expression patterns of *CmSPLs* and *CmmiR156* were also affected by GA treatment. During floral bud development, GA treatment significantly increased the expression level of *CmmiR156* and decreased the expression of the target gene *CmSPL6*/*CmSPL9*/*CmSPL16*. Some nontarget genes, such as *CmSPL7*/*CmSPL14*/*CmSPL18*, showed increased expression levels during some stages of floral bud development. The similar expression pattern was also found in *Phyllostachys edulis* and upland cotton [49,50], suggesting that the expression of the *SPL* genes may be regulated by GA and indicates that GA can indirectly affect the expression of *SPL* genes by regulating miR156 or directly regulate the expression level of *SPLs* by affecting the interaction between DELLA and SPL proteins.

## 4. Materials and Methods

### 4.1. Plant Materials

The Chinese chestnut cultivar “Yanshanhongli” was used in this study. Samples of different organs, including mixed floral bud, female flower, stigma, ovary, staminate catkin, male floral cluster, stamen, and leaf, were collected from 18-year-old Yanshanhongli plants in the Chestnut Experiment Station in the Huairou District of Beijing, China. The stigma and ovary samples were collected from unpollinated female flowers by removing the catkins and isolating the female flowers via bagging. All samples were cleaned after collection and immediately frozen in liquid nitrogen and stored at −80 °C until RNA extraction.

### 4.2. Exogenous Hormone Application

A stock solution containing 1 g of GA_3_ and 10 g of PP_333_ dissolved in 50 mL of 95% alcohol was diluted with distilled water to a final volume of 10 L. The final concentrations of GA_3_ and PP_333_ were 100 mg/L and 1000 mg/L, respectively. The control treatment involved the spraying of distilled water containing an equal amount of ethanol. Ten chestnut trees of similar growth were used for each treatment. The first spraying was performed prior to floral bud sprouting on 20 March 2017, and the trees were then sprayed three more times at 1-week intervals. In the middle of June, 40 branches at different positions were randomly selected from 10 chestnut trees in each treatment, and the total number and length of catkins of mixed inflorescences, male inflorescence, and female flowers in each sub-branch were investigated. SPSS 22.0 software was used for statistical analysis.

### 4.3. Identification of *CmSPL* Genes

Chinese chestnut gene and protein sequences were obtained from the Chinese chestnut genome project (https://www.ncbi.nlm.nih.gov/bioproject/527178), and two methods were used to identify *SPL* genes in Chinese chestnut. First, the hidden Markov model (HMM) profile of the SBP domain (pfam03110) was obtained from the Pfam website (http://pfam. xfam.org/) [51] and was employed as the query for the identification of all possible *SPL* genes in Chinese chestnut. Additionally, the *Arabidopsis* information resource database (http://www.arabidopsis.org/) was used to search the amino acid sequence of *AtSPLs*, and the sequence of *PtSPLs* was obtained from previous studies [18]. Then, the *AtSPL* and *PtSPL* sequences were used as queries for BLASTP searches against the Chinese chestnut genome database. All of the candidate *SPL* genes identified using the two methods were downloaded from the Chinese chestnut genome website. The reliability of these candidate sequences was verified using the Pfam (http://Pfam.sanger.ac.uk/) and NCBI databases (http://www.ncbi.nlm.nih.gov/), and the sequences lacking SBP domains were rejected.

### 4.4. Bioinformatic Analyses of *CmSPL* Genes

The amino acid sequence length, molecular weight, and isoelectric point of the candidate protein were calculated using the ExPASy website tool (http://web.expasy.org/protparam/). Conservative motifs were identified using the online tool MEME (http://meme-suite.org/tools/meme) with parameter settings referenced to previous studies [18], using the psRNATarget tool (http://plantgrn.noble.org/v1_psRNATarget/) to predict miiR156-targeted *SPL* genes.

### 4.5. Phylogenetic Analysis

The SBP domain sequences were obtained from SPL protein sequence alignment analysis using NCBI. A phylogenetic analysis was performed with reference to the MEGA 7.0 tool used by Kumar et al. [52].

### 4.6. RNA Isolation

The total RNA from tissues or organs of *C. mollissima* was extracted using the Plant RNA Kit (OMEGA, Norcross, GA, USA) according to the manufacturer’s instructions. The quality and quantity of total RNA were evaluated by agarose gel electrophoresis and using a Nanodrop One spectrophotometer (Thermo Scientific, Madison, NY, USA), respectively, and the elimination of genomic DNA through treatment with RapidOut DNA Removal Kit (Thermo Scientific, Vilnius, Lithuania).

### 4.7. Expression Analysis of *CmSPL* Genes

The cDNA was synthesized using M-MLV reverse transcriptase (Takara, Dalian, China). qRT-PCR analysis was performed by the Light Cycler96 Real Time PCR System (Roche Diagnostics GmbH, Mannheim, Germany) with SYBR Green (Takara, Dalian, China). The gene-specific primers were designed using Beacon designer software and are listed in Table A1. Each reaction system and procedure of qRT-PCR was based on the study of Lu et al. [53]. All qRT-PCR experiments were performed in three biological replicates. The relative expression levels were calculated using the 2−Δ*C*t method and *CmActin* was employed as the reference gene.

### 4.8. Expression Analysis of *CmmiR156*

Briefly, 2 μg of total RNA treated with DNase was used for first-strand cDNA synthesis using the *TransScript*^®^ miRNA First-Strand cDNA Synthesis SuperMix Kit (TransGen Biotech, Beijing, China). In the qRT-PCR analysis of miR156, *U6* was used as the reference gene. The program parameters were set as mentioned above, and the primers are listed in Table A1.

### 4.9. RLM-RACE

To identify the *CmmiR156* cleavage sites of the *CmSPL* transcripts, a modified 5’RLM-RACE program was used with the First Choice RLM-RACE kit (Invitrogen, Vilnius, Lithuania) following the method used by Li et al. [19]. Specific primers were designed based on predicted cleavage sites (Table A1), then the 5′ RACE PCR product was gel-purified and cloned into the pMD-19 T vector (Takara, Dalian, China), and 10 clones were randomly selected and sequenced.

## 5. Conclusions

In this study, we identified 18 *SPL* genes in the *C. mollissima* genome and analyzed their evolutionary relationships, gene structures, expression patterns, and potential functions. The development of male and female flowers in chestnut and the expression of some *SPL* genes were affected by GA treatment. The present work provides an important foundation for the future elucidation of the biological functions of *CmSPL* genes.

## Figures and Tables

**Figure 1 ijms-20-01577-f001:**
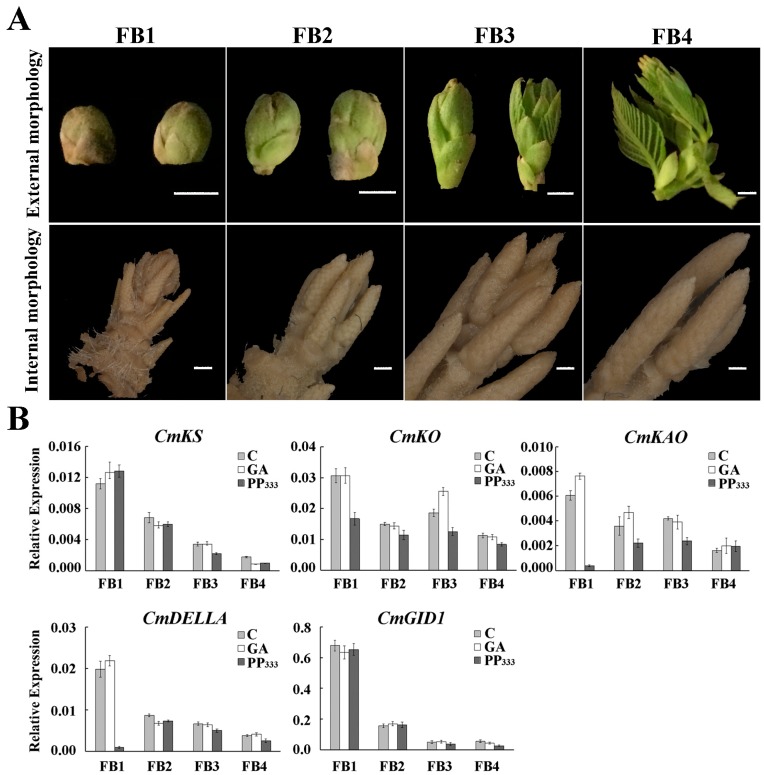
(**A**) Internal and external forms of floral buds at four time periods. The external morphology was imaged using an ordinary camera with a scale of 5 mm. The internal morphology was photographed using a stereomicroscope with a scale of 500 m. The Floral Bud 1 (FB1)–FB4 samples (corresponding to the four different time periods) were collected on 4, 10, 17, and 24 April 2017, respectively. (**B**) Expression profiles of key genes in gibberellin (GA) synthesis and signaling pathways at different floral bud stages under the three treatments. The *ent-kaurene synthase* (*KS*), *ent-kaurene oxidase* (*KO*), and *ent-kaurenoic acid oxidase* (*KAO*) genes are involved in the GA synthesis pathway, and the *DELLA* and *GA-insensitive drawf1* (*GID1*) genes are involved in the GA signaling pathway. The gray column represents the blank control, the white column represents the GA treatment, and the black column represents the PP_333_ treatment.

**Figure 2 ijms-20-01577-f002:**
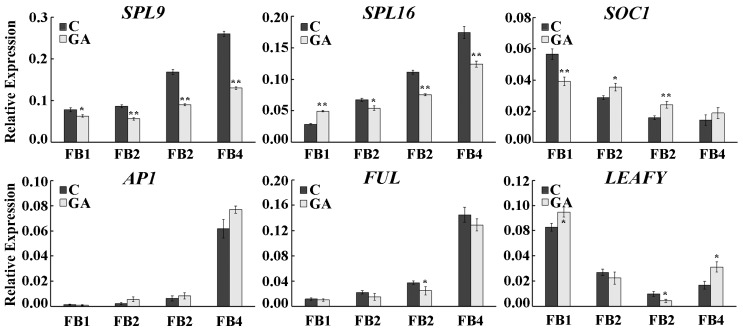
Expression patterns of *SPL9*, *SPL16*, *SOC1*, *AP1*, *FUL*, and *LEY* in floral buds at different stages under GA treatment. The black column represents the blank control, and the gray column represents the GA treatment. Asterisks indicate significant differences according to a Student’s *t*-test (* *p* ≤ 0.05; ** *p* ≤ 0.01).

**Figure 3 ijms-20-01577-f003:**
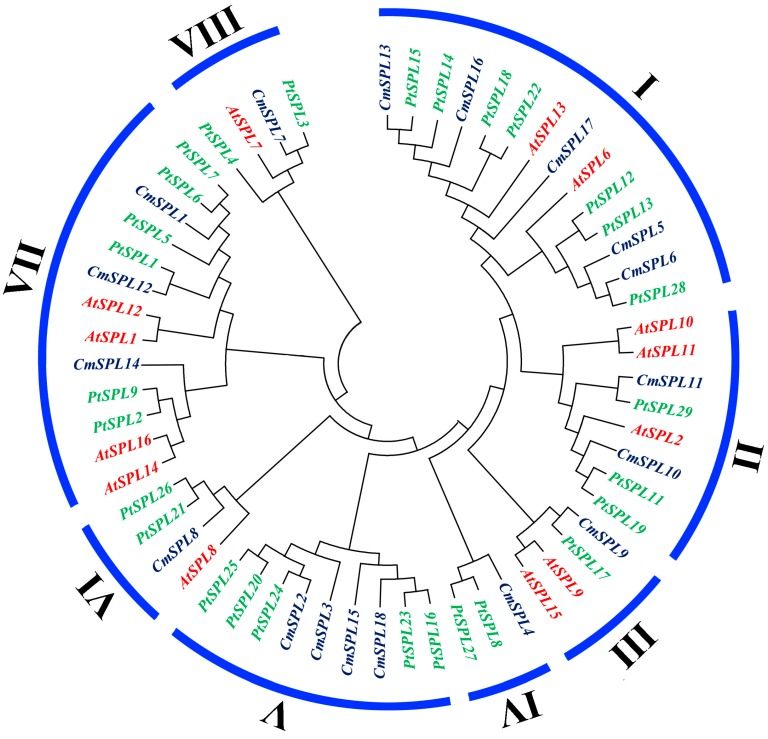
Phylogenetic tree of 59 *SPLs* in *C. mollissima*, *Arabidopsis thaliana*, and *Populus trichocarpa*. The phylogenetic tree was constructed based on the SBP domain using the neighbor-joining (NJ) method with MEGA 7. The number on the branch indicates the bootstrap value, and the *SPL* genes in the same species are represented with the same colors: purple, *C. mollissima*; red, *A. thaliana*; and green, *P. trichocarpa*.

**Figure 4 ijms-20-01577-f004:**
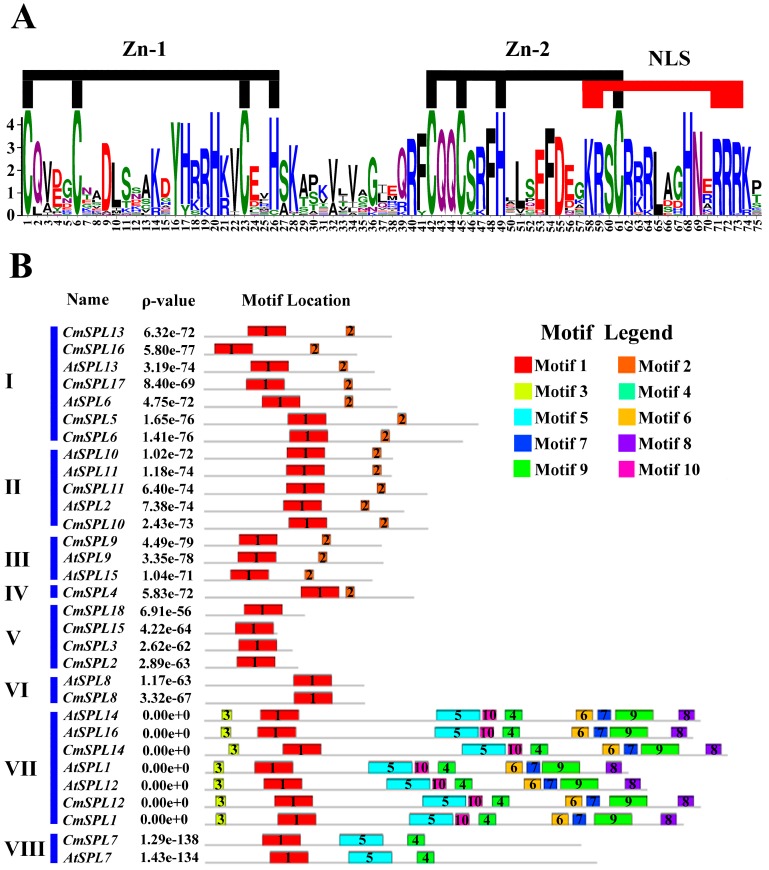
Conserved motif analysis. (**A**) Sequence logo of the SBP domain of *CmSPLs*. The overall height of each stack represents the degree of conservation at this position, whereas the height of the letters within each stack indicates the relative frequency of the corresponding amino acids. (**B**) Distribution of conserved motifs in *CmSPLs*. The motifs represented with boxes were identified using the MEME online tool. The numbers in boxes 1–10 represents motifs 1–10, respectively. The position and length of each colored box represents the actual motif size.

**Figure 5 ijms-20-01577-f005:**
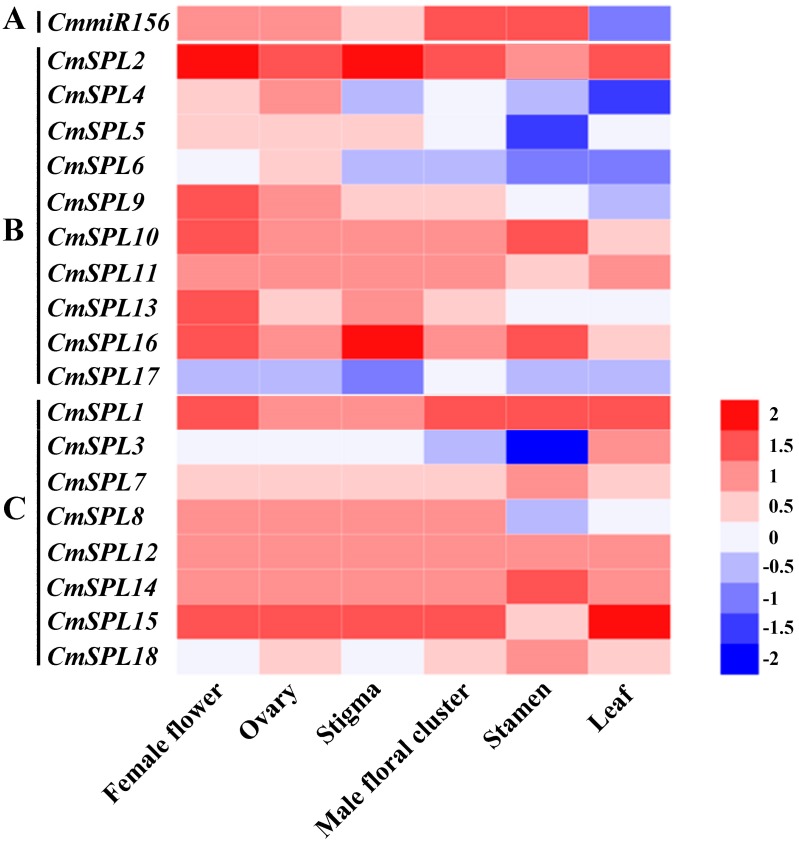
Expression profiles of the Chinese chestnut *CmSPL* genes in female flower, male floral cluster, leaf, ovary, stigma, and stamen. The heat map was generated based on the mean values from the qRT-PCR data obtained from the different samples. The red and blue colors represent higher and lower expression, respectively.

**Figure 6 ijms-20-01577-f006:**
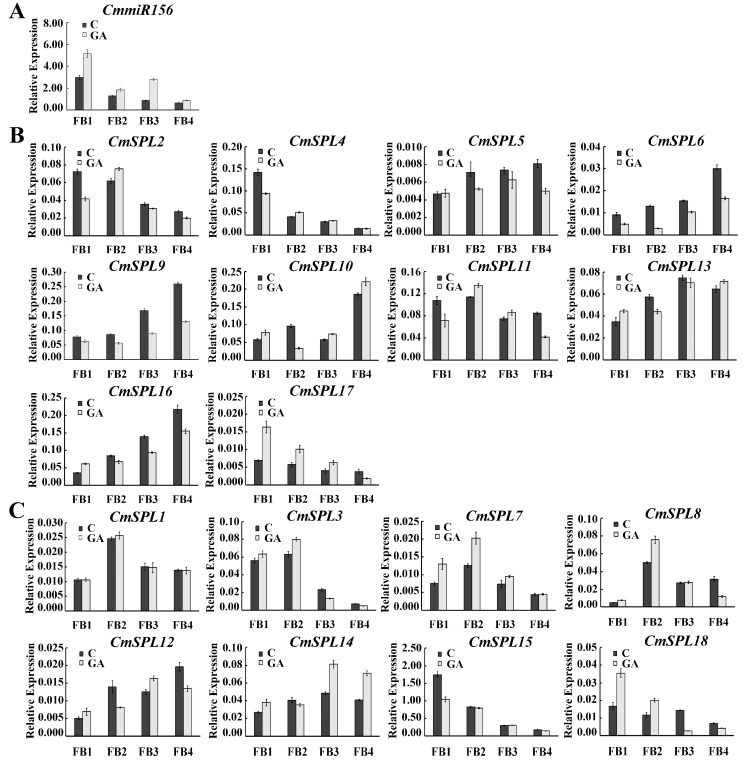
Expression profiles of mature *CmmiR156* and *CmSPLs* in floral buds of *C. mollissima* at different stages under GA treatment. (**A**) Expression patterns of *CmmiR156*. (**B**) Expression patterns of *CmmiR156*-targeted *CmSPLs*. (**C**) Expression patterns of *CmmiR156*-nontargeted *CmSPLs*. Floral Bud 1 (FB1)–FB4 represent the floral buds at four different periods (4, 10, 17, and 24 April 2017, respectively).

**Figure 7 ijms-20-01577-f007:**
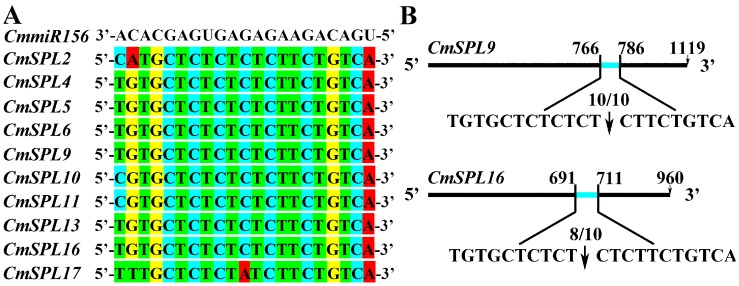
Identification of *CmmiR156*-guided cleavage products of target genes. (**A**) Sequence of the miR156 complementary region in *CmSPL* genes. (**B**) Sites for the cleavage of *CmSPL9* and *CmSPL16* by *CmmiR156*. The black arrow indicates the cutting site.

**Table 1 ijms-20-01577-t001:** Multiple test (LRS) results of the effects of various growth regulators on the number of male and female flowers.

Species	Concentration (mg·L^−1^)	Female Flower Cluster /Branch	Number of Mixed Inflorescences /Branch	Mixed Inflorescence Length (cm)	Number of Male Inflorescence /Branch	Ratio of Female Flower Clusters to Mixed Inflorescence Number	The Ratio of the Number of Female Flowers to the Number of Male Inflorescences	Ratio of Mixed Inflorescence Number to Male Inflorescence Number
C	0	2.71 ± 1.62b	2.11 ± 0.93a	16.33 ± 2.61a	8.44 ± 3.43b	1.5 ± 0.40a	0.47 ± 0.27a	0.34 ± 0.15a
GA	100 mg/L	2.36 ± 0.50b	2.14 ± 0.53a	12.65 ± 2.34b	11.14 ± 2.98a	1.14 ± 0.31b	0.22 ± 0.06b	0.21 ± 0.07a
PP_333_	1000 mg/L	3.70 ± 1.42a	2.40 ± 0.84a	17.40 ± 2.94a	8.80 ± 1.87b	1.58 ± 0.34a	0.46 ± 0.25a	0.30 ± 0.15a

Different lowercase letters (a and b) in each column indicate significantly differences between samples at 5% level.

**Table 2 ijms-20-01577-t002:** Gene feature and classification of Squamosa promoter-binding protein-like (*SPL*) in *Castanea mollissima*.

Gene Name	CDS (bp)	Peptide (aa)	Mw (Da)	pI	Ai	miR156 Target
*CmSPL1*	3000	999	110899.70	5.77	84.41	No
*CmSPL2*	582	193	21864.54	9.39	51.50	Yes
*CmSPL3*	549	182	20368.72	8.73	51.43	No
*CmSPL4*	1314	437	48311.02	7.05	57.41	Yes
*CmSPL5*	1722	573	62793.15	6.66	57.87	Yes
*CmSPL6*	1623	540	60015.14	6.15	68.24	Yes
*CmSPL7*	2358	785	87807.21	5.72	76.28	No
*CmSPL8*	1026	341	37999.46	8.98	44.75	No
*CmSPL9*	1119	372	39809.28	9.37	47.55	Yes
*CmSPL10*	1410	469	52259.35	8.65	52.47	Yes
*CmSPL11*	1398	465	51284.57	8.03	58.11	Yes
*CmSPL12*	3108	1035	114576.19	7.32	84.38	No
*CmSPL13*	1176	391	42863.75	8.80	62.58	Yes
*CmSPL14*	3276	1091	120636.02	7.86	74.28	No
*CmSPL15*	456	151	17311.85	6.71	29.74	No
*CmSPL16*	960	319	34804.61	9.15	60.19	Yes
*CmSPL17*	1173	390	43054.10	8.47	69.00	Yes
*CmSPL18*	624	207	23511.61	9.32	53.24	No

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
