# Peer review of "Roles of the GA-mediated SPL Gene Family and miR156 in the Floral Development of Chinese Chestnut (Castanea mollissima)"

_ijms, 2019, doi:10.3390/ijms20071577_

Round 1

Reviewer 1 Report

Please correct the comments in pdf

Author Response

Dear Editors and Reviewers:
  Thank you for your letter and for the reviewers comments concerning our manuscript entitled “Roles of the GA-mediated SPL gene family and miR156 in the floral development of Chinese chestnut (Castanea mollissima)”. ID: ijms-467172. Those comments are all valuable and very helpful for revising and improving our paper, as well as the important guiding significance to our researches. We have studied comments carefully and have made correction which we hope meet with approval. The main corrections in the paper and the responds to the reviewers comments are as following:

1. Reduce the  font maybe then it will come one line.

Answer: Yes, we have made changes in the manuscript. In addition, other modifications have been detailed in the manuscript.

Reviewer 2 Report

Minor questions:

1.     Lines 121-123: table1 shows a decreased ratio of female to male after GA implement, which contracts the statements here.

2.     Line 156, it seems that FUL is also induced during development. Also, for figure 2, how about gene expression under PP333 treatment? Does it support GA regulation?

3.     Figure 3: phylogeny should be implemented with a sophisticated approach, such as ML or Bayes. NJ does not necessary to represent evolutionary distance. The group VIII shouldn’t include Arabidopsis genes because they are not forming a MONOCLADE.

4.     Lines 237-238: the conclusion is too arbitrary, similar expression pattern does not warrant functional redundancy.

Author Response

Dear Editors and Reviewers:
  Thank you for your letter and for the reviewers’ comments concerning our manuscript entitled “Roles of the GA-mediated SPL gene family and miR156 in the floral development of Chinese chestnut (Castanea mollissima)”. ID: ijms-467172. Those comments are all valuable and very helpful for revising and improving our paper, as well as the important guiding significance to our researches. We have studied comments carefully and have made correction which we hope meet with approval. The main corrections in the paper and the responds to the reviewer’s comments are as following:

1.     Lines 121-123: table1 shows a decreased ratio of female to male after GA implement, which contracts the statements here.

Answer: I am very sorry, this is a mistake of me and has been revised in the article.

2.     Line 156, it seems that FUL is also induced during development. Also, for figure 2, how about gene expression under PP333 treatment? Does it support GA regulation?

Answer: In our study, we performed Student’s t test using SPSS 22.0 and found that there was no significant difference in the expression level of the FUL gene compared to the control group under GA treatment. Therefore, we did not analyze the expression level under PP333 treatment. 

Figure 2. Expression patterns of SPL9, SPL16, SOC1, AP1, FUL and LEY in floral Bud at different stages under GA treatment. The black column represents the blank control, and the gray column represents the GA treatment. Asterisks indicate significant differences according to a Student’s t Test (*P≤0.05;**P≤0.01)

3.     Figure 3: phylogeny should be implemented with a sophisticated approach, such as ML or Bayes. NJ does not necessary to represent evolutionary distance. The group VIII shouldn’t include Arabidopsis genes because they are not forming a MONOCLADE.

Answer: Yes. We have modified the phylogenetic tree according to your suggestion. The results are as follows, and we have also modified Figure 4 and the corresponding description in the text.

Figure 3. Phylogenetic tree of 59 SPLs in C. mollissima, A. thaliana, and P. trichocarpa. The phylogenetic tree was constructed based on the SBP domain using the neighbor-joining (NJ) method with MEGA 7. The number on the branch indicates the bootstrap value, and the SPL genes in the same species are represented with the same colors: purple, C. mollissima; red, A. thaliana; and green, P. trichocarpa.

4.     Lines 237-238: the conclusion is too arbitrary, similar expression pattern does not warrant functional redundancy.

Answer: Yes, we have made changes in the manuscript. In addition, other modifications have been detailed in the manuscript.
